# Application of Bagged Copula-GP: Confirming Neural Dependency on Pupil Dilation

Maximilian Walden            *M.Walden@sms.ed.ac.uk*
*University of Edinburgh*

Reviewed on OpenReview: *https://openreview.net/forum?id=hQIs1CHF6H*

## Abstract

Advancements in recording techniques have enabled the ability to record thousands of neurons simultaneously, shifting the needs within the field of computational neuroscience to that of powerful computational and statistical techniques. Copula-GP is a recently developed state-of-the-art parametric mutual information estimator found to outperform other novel non-parametric methods when utilized on highly dimensional data. Here, we utilized Copula-GP together with *Gaussian Process Factor Analysis* (GPFA) to investigate the information interaction between neuronal processes within the visual cortex of live mice and pupil dilation. We found usage of GPFA as a preprocessing step to Copula-GP was an effective means of investigating neuronal dependence, allowing flexibility in analysis and finding results in agreement with prior literature. We additionally extended Copula-GP with a bagging framework, allowing for the aggregation of model estimations and allowing for more accurate estimation accuracy and representation of dependency shape. We validated our bagging algorithm on simulated data sampled from known distributions, and utilized bagged Copula-GP with GPFA on said neuronal data to find results in agreement with baseline Copula-GP but with more stability.

## 1 Introduction

Computational Neuroscience aims to answer how cognition, behavior, and learning are encoded in the brain through the use of machine learning and statistical methods on large amounts of neuronal data (Blundell et al., 2018; McDougal et al., 2017; Vu et al., 2018), with answers to these questions having applications in fields such as prosthetics, medicine, and machine learning (Jangid et al., 2021). Some of the primary questions on this frontier include: How is information encoded in populations of neurons, and how is this information related to behaviors observed (Kudryashova et al., 2022)?

As advances in recording techniques have allowed for the recording of hundreds to thousands of neurons at once (Jun et al., 2017; Dombeck et al., 2007), the bottlenecks keeping researchers from answering these central questions in neuroscience have shifted away from data-related issues to the purely computational and statistical (Vu et al., 2018). Answering such questions with machine learning comes down to analysis of the intricate highly-dimensional multivariate dependencies (i.e. dependencies and correlations between many random variables, many of which have different distributions) in recorded neuronal and behavioral data, much of which varies across spatial and temporal dimensions (Kudryashova et al., 2022; Jun et al., 2017; Glaser et al., 2019; Quian Quiroga & Panzeri, 2009). This becomes especially challenging once one considers how different behaviors can occur on a scale of hours or even days, whereas related neuronal activity can occur on a scale of milliseconds (Pakan et al., 2018; Dombeck et al., 2007; Mathis et al., 2018), even more so as more variables are included in the data and the dependencies become even more computationally intense to analyze accurately (the "curse of dimensionality") (Koppen, 2000). Thus, in order to properly analyze such copious amounts of intricate high-dimensional data, we need methods that are both resistant to high dimensionality as well as equipped with the ability to properly analyze dependencies between temporally-disparate variables in time series.

*Parametric Copula-GP model for analyzing multidimensional neuronal and behavioral relationships* (Kudryashova et al., 2022) aimed to fulfill this requirement via a novel framework for parametric estimation of multivariate distributions and mutual information through the use of *copulas*, a probabilistic structure adept at mutual information estimation. The core paper's contributions to the body of work includes the development of the Copula-GP python package, a package which allows for the training of copulas, model selection, and copula entropy extraction. Copula-GP was found to outperform other non-parametric mutual information estimators when predicting mutual information for a larger number of variables ($\geq 10$). This combined with it's robustness to the "curse of dimensionality" makes it well suited for analysis of highly dimensional neuronal data.

Here, we validated usage of Copula-GP together with *Gaussian Process Factor Analysis* (GPFA; a dimensionality reduction method robust to qualities of neuronal data through which we are able to extract latent variables), doing so through confirming statistical dependence of neuronal data extracted from mice on pupil dilation. Within the literature, there is very little use of GPFA with mutual information estimators, much less conditional parametric copulas on neuronal data (or on any data). This project aimed to fill that gap, with the goal of improving the interpretability of the model by capturing processes in the brain across groups of neurons as latent trajectories extracted from spike-train data and estimating the mutual information between latent trajectories $X_1, X_2, \cdots, X_N$ conditioned on a behavioral variable $Y$, as well as the information interaction $I(X_1 : X_2 : \cdots : X_N \leftarrow Y)$ measuring statistical dependence in $X_1, X_2, \cdots, X_N$ captured by $Y$.

We also aimed to extend Copula-GP with a method of ensemble model selection through the use of *bagging*, wherein we aggregate copula estimations of individual Copula-GP estimators fit on separate samples of data, validating the ensemble method via entropy estimation of a known distribution and culminating in usage of bagged Copula-GP on aforementioned real life neuronal data. To summarize, the contributions of this paper include:

- Validation of combining novel Copula-GP with GPFA (the combination of which we dub *Copula-GPFA* as shorthand), with the expected benefits of robustness to dimensionality and efficient, detailed analysis of neuronal dependence.

- Justification of possible improvement to the Copula-GP algorithm through the addition of bagging, confirmed via validation tests on synthetic data sampled from known copula distributions.

- Confirming dependencies between neuronal data from the visual cortex on pupil dilation through the use of bagged Copula-GPFA.

The code repository for work shown can be found at: `https://github.com/mwalden00/HonProj`.

## 2  Neuroscience Preliminaries

In general, if a system of neurons encodes the information that goes into behaviors or cognition then we expect uncertainty in neuron spikes to be captured in behaviors during recorded spike-trains, or in other terms a significant quantity of mutual information and/or information interaction between spike signals and behaviors (Gerstner et al., 2014). By extension, if we are able to accurately estimate mutual information then we should thusly be able to map systems of neurons to behaviors and cognition.

As stated in the introduction, computational neuroscience aims to answer questions surrounding the encoding of cognition and behavior in parts of the brain. Hurdles in answering such questions in the past stemmed from the difficulty of recording groups of neurons' behavior with certainty (Kudryashova et al., 2022), however we now have the technology needed for accurate single-neuron resolution spike train trial recordings in the form of *Neuropixel probes* (Steinmetz et al., 2021). As such, the hurdle of neuron-behavior-stimulus mutual information analysis is no longer centered on lack of data, moving instead to the need for accurate and efficient computational and statistical mutual information estimation techniques, as well as required robustness to dimensionality when a large number of individual neurons and behavioral variables are analysed (Kudryashova et al., 2022; Onken & Panzeri, 2016; Vu et al., 2018).

## 3 Related Work

There are many choice methods in the realm of neuronal mutual information estimation. Popular novel ones include Bias-Improved Kraskov-Stögbauer-Grassberger (BI-KSG) (Gao et al., 2018) and the Mutual Information Neuronal Estimator (MINE) (Belghazi et al., 2021), both of which are effective non-parametric methods. The Copula-GP method implemented in the core paper by (Kudryashova et al., 2022) is a parametric estimator found to generally outperform MINE and BI-KSG in mutual estimation accuracy on highly-dimensional data ($\geq 10$ dimensions), and as such is our chosen method of neuronal mutual information analysis. It is a method based on *copulas*, a multivariate distribution with uniform marginals representative of cumulative distributions and a history of applications ranging from scientific analysis of dependency (as they are generally used in computational neuroscience (Jenison & Reale, 2004)) to the more purely analytical and predictive (i.e. predicting the outcome of an insurance claim (Hu & O'Hagan, 2021)). We delve more into what copulas are and how they're useful in section 4.

Copulas have been used in mutual-information estimation and dependency analysis in a variety of *in vivo* and *in silicon* neuronal data, including spike data, 2-photon calcium imaging, and multi-modal neuronal datasets, and have been successfully used for both mutual information estimation and dependency shape analysis in neuronal data (Onken & Panzeri, 2016; Jenison & Reale, 2004; Shahbaba et al., 2014; Berkes et al., 2008). However, recent advances in GPU accelerated processing power (through the use of popular libraries such as *PyTorch* (Paszke et al., 2019) and *GPyTorch* (Gardner et al., 2018)) have allowed for further refinement of copula implementations and expanded the number of practical use cases, allowing for the implementation of such a method for high-dimensional neuronal data, like that of Copula-GP (Kudryashova et al., 2022).

## 4 Copulas: An Introduction

Copulas are multivariate distributions with uniform marginal distributions, which themselves usually represent marginal variables' cumulative distributions. A $d$-dimensional copula function $C(u_1, u_2, \ldots, u_d) : [0,1]^d \to [0,1]$ is defined as a *cumulative distribution function (CDF)* of a vector on the unit hyper-cube $[0,1]^d$ with uniform marginals $\mathcal{U}_{[0,1]}$ (Kudryashova et al., 2022; Safaai et al., 2018). Variables of non-uniform distribution are "attached" to the marginals through the use of CDFs (which by definition map to the interval $[0,1]$), creating a copula as a joint CDF (Nelsen, 2006). *Sklar's theorem* states that for a $d$-dimensional random vector $\mathbf{X} = \{X_1, X_2, \cdots, X_d\}$ and it's CDF $F_{\mathbf{X}}$ with marginals $F_1, F_2, \cdots, F_d$, there exists a copula $C$ such that $\forall \mathbf{x} \in \mathbb{R}^d$, $\mathbf{x} \sim \mathbf{X}$,

$$F_{\mathbf{X}}(x_1, x_2, \ldots, x_d) = C(F_1(x_1), F_2(x_2), \ldots, F_d(x_d)), \quad x_i \in \mathbb{R}. \tag{1}$$

(Safaai et al., 2018; Van Vliet, 2023) What is most valuable about this construction is the allowance for the random variables $X_1, X_2, \cdots, X_d$ to take *any* distribution, allowing for meaningful dependency and mutual information analysis between variables of different probabilistic distributions (Safaai et al., 2018). We can also condition the copula on some continuous variable $y$, allowing for the parametrization of the copula for a variable like time, phase, other marginals, etc (Kudryashova et al., 2022):

$$F_{\mathbf{X}}(x_1, x_2, \ldots, x_d | y) = C(F_1(x_1|y), F_2(x_2|y), \ldots, F_d(x_d|y)|y), \quad x_i, y \in \mathbb{R}. \tag{2}$$

To scale against higher dimensions, copulas can take a *vine copula construction*, which factorize multivariate distributions into conditional distributions modelled as singular bivariate copulas (Onken & Panzeri, 2016; Mitskopoulos et al., 2022). The *canonical* vine-construction factorises a multivariate distribution in this was as

$$f_{\mathbf{X}}(x_1, \cdots, x_d) = \prod_{k=1}^{d} f(x_k) \prod_{j=1}^{d-1} \prod_{i=1}^{d-j} C_{j,i|1,\cdots,j-1} \left( F(x_j | x_1, \cdots, x_{j-1}), F(x_{i+j} | x_1, \cdots, x_{j-1}) \right). \tag{3}$$

A divide-and-conquer approach such as this allows us to mitigate the curse of dimensionality given sufficient performance in the bivariate case Mitskopoulos et al. (2022); Kudryashova et al. (2022). For flexible paramerization of a copula, *Gaussian Process (GP)* can be used for parametrization. GP in essence attempts to model a relationship between random variables within a stochastic process such that all collections of those

variables possess a gaussian distribution, in effect modelling a distribution of functions from which one can sample Rasmussen & Williams (2005). GP can be used to estimate a distribution $\mathbf{f}$ from which a function $f \sim \mathbf{f}$ describing parameterization over a domain $[0, 1]$ is sampled, which can then be used in conjunction with GPLink functions to parameterize copula variants (see table 2 in the appendix for precise definitions of GPLinks) (Kudryashova et al., 2022; Hernández-Lobato et al., 2013). To express this mathematically, we redefine individual copulas to take the approximate form

$$F_{\mathbf{X}}(x_1, \cdots, x_d | y') = C(F_1(x_1 | \theta_1(y')), F_2(x_2 | \theta_2(y')), \cdots, F_d(x_d | \theta_d(y')) | \theta'(y')), \tag{4}$$

where $\theta_1, \cdots, \theta_d, \theta' : \mathbb{R} \to \text{dom(y)}$ are functions mapping some continuous variable $y'$ onto the domain of the conditioning parameter $y$ representing the relationships between $x_1, \cdots, x_d$ and $y'$, often called *GPLink* functions (Kudryashova et al., 2022; Schulz et al., 2018).

Finally, we define *copula entropy* as

$$H_C(\mathbf{X}) = - \int_C c(\mathbf{u}) \log c(\mathbf{u}) d\mathbf{u}, \tag{5}$$

where $c(\mathbf{u})$ represents the probability density of the copula at $\mathbf{u}$

$$c(\mathbf{u}) = \frac{\partial^d C}{\partial u_1 \partial u_2 \ldots \partial u_d}. \tag{6}$$

(Ma & Sun, 2011; Jenison & Reale, 2004) As proven by Ma & Sun (2011), copula entropy is equivalent to the negative mutual information of the marginals, i.e. $I(\mathbf{X}) = -H_C(\mathbf{X})$. In other words, through accurate estimation of a target distribution's copula we are able to accurately estimate the mutual information between that distributions' marginal variables.

## 5 Copula-GP

The Copula-GP package was implemented in Python and deployed in Kudryashova et al. (2022), and is one of the few (if not only) practical implementations of a GP-treated copula vine model designed with GPU based acceleration of computation via *Pytorch* (Paszke et al., 2019). It is also the first of such models designed with use on neuronal data in mind, and as such literature on the uses of GP vine copulas on modelling neuronal data is heavily limited, with the main source for such claims being the core paper. In simplest terms, the package serves as a framework for GP copula model parameter estimation, copula model selection, and model deployment as a vine, with additional built in visualization and entropy extraction utility. The core paper also utilizes comparison to other popular methods of neuronal mutual information extraction, yielding superior results in highly-dimensional data. **Any subsequent claim surrounding the Copula-GP package without a citation attached in this section comes courtesy of (Kudryashova et al., 2022)**.

The vine construction implemented in Copula-GP used copula building blocks from five distinct families: independence, Gumbel, Gaussian, Frank, and Clayton copulas, with the independence copula preferred for independent variables (see figure 6). The factorization of the vine is that of the C-vine described by equation 3, with accommodations made for GP-parameterization. To fully capture the tail dependencies and negative correlations in relationships between marginal distributions, *mixed copulas* were utilized. In the core paper, these are defined as

$$C_{mixed}(\mathbf{X} | Y) = \sum_{j=1}^{K} \phi_j(Y) C_j(\mathbf{X} | \theta_j(Y)), \tag{7}$$

where $K$ is the number of elements, $\phi_j$ is the *concentration* of the $j$th copula ($c_j$), and $\theta_j$ is the $j$th copula's parameter GPLink. The GPLink for each copula is determined by its copula variant, as shown in figure 2, with $\theta$ being defined by GPLink($f$), where $f$ is sampled from $\theta \sim \mathcal{N}(\mu, K_\lambda(X, X))$ (the choice of GPLink

depends on the kind of copula; see 2). GP is also utilized to parameterize the concentrations $\phi_j$, which are defined as

$$\phi_j = (1 - t_j) \prod_{m=1}^{j-1} t_m, \quad t_m = \Phi\left(\tilde{f}_m + \Phi^{-1}\left(\frac{M - m - 1}{M - m}\right)\right), \quad t_M = 0, \tag{8}$$

where $\Phi$ is the CDF of a standard normal distribution and $\tilde{f}_m$ is sampled from $\tilde{\mathbf{f}}_\mathbf{m} \sim \mathcal{N}(\tilde{\mu}_m, \ \tilde{K}_{\tilde{\lambda}_m}(Y, Y))$. This gives us $2M - 1$ sets of hyper parameters to estimate, $\{\lambda\}_M$ kernel hyperparameters for each GPLink $\theta$ and $\{\tilde{\lambda}\}_{M-1}$ kernel hyperparameters for each concentration function $\phi$, estimated via the methods described in section E.1.

# 6 Entropy and Mutual Information Estimation

As Copula-GP models a copula distribution, possible states of variables can be sampled from it. As such, the joint mutual information $I(X_1 : X_2 : \cdots : Y) = H_C(\mathbf{X}) - H_C(\mathbf{X}|Y)$ between parameter $Y$ and multivariate distribution $\mathbf{X} = \{X_1, X_2, \dots\}$ can be derived from $C(\mathbf{X}|Y)$ (the C-vine copula Copula-GP estimates). Copula-GP possesses built-in capabilities for computing mutual information directly via the above equation, however doing so requires computation of nested integrals and as such is computationally intensive; in testing, when attempting to extract the mutual information of complex distributions we found our machine quickly exhausted GPU memory when utilizing a single NVidia 2080Ti. We choose to find the MC estimate (Robert & Casella, 1999) of $H_C(X)$ (which is also utilized in the core paper), which involves sampling from the distribution $C(X|\hat{y}_i)$ with parameterization in $S$ random values $\hat{y}_i \sim \mathcal{U}(0, 1)$ and estimating the conditional entropy as the mean

$$H_C(X|Y) = \int_{\text{dom}(y)} H_C(X|y_i)dy_i \approx \frac{1}{S}\sum_{i=1}^{S} H_C(X|\hat{y}_i). \tag{9}$$

Similarly, we estimate the unconditional entropy as the mean entropy of the system found when fed true values of the parameterizing variable $y_i \in [0, 1]$, i.e

$$H_C(X) = \mathbb{E}_y(H_C(X|y)) \approx \frac{1}{S}\sum_{n=1}^{S} H_C(X|y_n). \tag{10}$$

Due to the law of large numbers, as $N$ goes to infinity both approximations become more accurate. To find the point-wise copula entropy values $H_C(X|y_i)$, we utilize Copula-GP's implemented entropy extraction methods

# 7 Extension of Copula-GP: Bagging

*Bagging* is a bootstrapping technique wherein multiple "weak" models are trained and their outcomes aggregated in some way to create a "stronger" model, typically reducing variance and increasing model bias. We propose a formal justification of copula bagging, taking inspiration from the "Random Forest" algorithm for bagged regression (Breiman, 2001); we assume the existence of some true copula function $C$ describing a multivariate distribution $X$. We take $N$ samples $\{S_1, S_2, \dots, S_N\}$ and get a copula $C^{(n)}$ representing the best copula fit possible on an individual sample $S_n$. We then come to a final estimate via a mean aggregation of our estimates. Suppose $C^{(n)}$ is a C-vine estimate, and let $C_{i,j}^{(n)}$ represent the $i$-th copula in the $j$-th layer of the $n$-th estimate. We can create a mean mixture copula $\hat{C}_{i,j}$ representing the final estimate for the corresponding true copula $C_{i,j}$ as $\hat{C}_{i,j} = \frac{1}{N}\sum_{n=1}^{N} C_{i,j}^{(n)}$. As $N \to \infty$, by the law of large numbers (and the fact that copulas are distributions) we find $\hat{C}_{i,j} \to C_{i,j}$, and so $\hat{C} \to C$. In other words, by fitting copulas on individual spike-train trials and taking the aggregate, we estimate a copula that more closely aligns with the true probabilistic relationship than those aggregated.

## 7.1 Implemented Weighted Aggregation Methods

A simple unweighted mean is appropriate in the infinite case. In practice however, we often find that a naive average is insufficient when examining a finite number of samples; suppose we have finite $N$ uniform length

samples $S_n$ and thus have a set of observations $S = \{S_1, S_2, \ldots, S_N\}$, fitting corresponding copula estimates $\hat{C}_n$. If the $n$-th sample is an outlier sample and so $\hat{C}_n$ does not resemble the true copula $C_n$, we naturally should account for that. As such, in addition to a naive average, we may choose to weigh models based on various Bayesian criterion. We utilized a Bayesian model aggregation approach utilized by S. Hu *et. al* (Hu & O'Hagan, 2021), which aggregated copulas via *Bayesian Information Criterion* (BIC). The BIC value for the $n$-th estimated copula is given as

$$\text{BIC}_\text{n} = -2\log\mathcal{L}(\hat{C}_n|X, \Theta) + p_n \log|X|, \tag{11}$$

where $\log\mathcal{L}(\hat{C}_n|X,\Theta)$ is the log-likelihood of copula $\hat{C}_n$ under observations $X$ and estimated model parameters $\Theta$, and $p_n$ is the total number of estimated parameters of $\hat{C}_n$ (in the case of Copula-GP, each bivariate copula estimated is a mixture copula and so possesses mixing and dependence parameters for each copula mixed; if the mixture is a singleton mixture, we only count dependence parameter. Note independence copulas have no dependence parameter). BIC essentially rewards model log-likelihood with penalty in number of parameters scaling with sample-size (Schwarz, 1978; Hu & O'Hagan, 2021), with small BIC values are being preferred (as negative as possible). We arrive at weights for the $n$-th model as

$$W_{n,\text{BIC}} = \frac{\exp(-\frac{1}{2}BIC_n)}{\sum_m \exp(-\frac{1}{2}BIC_m)}. \tag{12}$$

(Hu & O'Hagan, 2021) We also utilize the *Akaike Information Criterion* (AIC) for comparison, which is given for the $n$-th copula estimation as $\text{AIC}_\text{n} = -2\log\mathcal{L}(X|\hat{C}_n,\Theta) + 2p_k$ and extract weights $W_{n,\text{AIC}}$ accordingly via equation 12, replacing $\text{BIC}_n$ with $\text{AIC}_n$. The AIC is similar to the BIC in that it rewards log-likelihood but has a more relaxed penalty in number of parameters. If a more complex copula matches the true distribution, this can lead to possible improvements in aggregation accuracy, but can also lead to over-fitting via over-parameterization (Zhou, 2021) (as we are already utilising mixtures of copulas with a high ceiling in number of parameters, we expect the latter to be true). We additionally include dynamic weighting of estimates via calculating the above information criteria and their respective weights point-wise as opposed to setting weights to be constant over the observations X; we call point-wise aggregation *dynamic bagging* and constant weighting *static bagging*. We validate our bagging methods with different aggregation methods on data samples from randomly generated bivariate copulas in section 9.

To aggregate C-vines, we bag one layer at a time; we first bag copulas in the current layer via our selected method and gather cumulative conditional probabilities as pseudo-observations via the bagged copulas' conditional cumulative distribution functions (ccdfs). We then utilize them for BIC, AIC, and $R^2$ calculations in the next layers' bagging process, and in doing so effectively propagate the previous layers' weightings to the next layer. As subsequent bagged copulas will have worse BIC, AIC, and $R^2$ calculations if their corresponding C-vines' previous layers were found to be bad fits, we in effect prioritize relationships with weaker conditioning using this method.

Copula-GP estimates a parametric C-vine along a given 1-D parameterization input $X$, modeling each individual copula as a mixture and getting estimated dependence parameterizations $\{\theta_{i,j}(X)\}$ and estimated mixing parameterizations $\{\phi_{i,j}(Y)\}$ for each of $i$ mixture copulas (with each mixture copula being a mix of $j^{(i)}$ copula variants). As such, our actual implemented bagging procedure is to aggregate along these estimated parameters for each unique copula variant in each mixture (we count each rotation variant of a Clayton or Gumbel copula as unique; this algorithm is outlined more formally in the appendix, see algorithm 1). In addition, we implemented model selection for each C-vine we aggregate, allowing for the bagged vine to contain copula mixtures possibly unexamined in the heuristic based model selection process. The C-vine aggregation process requires aggregation of copulas by layer and storing the cumulative distributions of each copula via the bagged copula ccdfs for weight extraction in the next layer.

## 8    Gaussian-Process Factor Analysis

A very common technique in the realm of dimension reduction for highly dimensional neuronal data is *Gaussian-Process Factor Analysis* (GPFA), which factors a matrix of observations $X$ into a product of a loading matrix $\Psi$ and a scoring matrix $\Theta$ as $X = \Psi\Theta$ (the *Factor Analysis* step) producing a low-dimensional

group of trajectories, and then smooths said data by means of fitting a GP (the *Gaussian Process* step) (Pirš & Štrumbelj, 2022). This operates under the assumption of a linear relationship between observations $X$ and latent trajectories $\Theta$ (utilizing $\Psi$ as a transformation matrix) (Yu et al., 2009), and models the GP with bias $d$ as

$$X = \Psi\Theta + d + \epsilon, \ \ \epsilon \sim \mathcal{N}(0, \sigma_\epsilon^2) \tag{13}$$

with parameters $(d, \Psi, \sigma_\epsilon^2)$. The Copula-GPFA process consists of GPFA applied to neuronal data to extract latent trajectory estimations $\hat{X}_1, \hat{X}_2, \ldots$, followed by fitting a Copula-GP C-vine estimator on neuronal trajectories. Doing so, we are able to efficiently extract mutual information estimates describing not just the information interaction between the examined part of the brain and some parameterizing variable, but the mutual information between the brain processes driving neurons instead of individual neurons themselves (Yu et al., 2009; Ma & Sun, 2011; Kudryashova et al., 2022), enhancing the interpretability of individual copulas estimated. In addition, by utilizing GPFA to reduce a large number of neurons to a more manageable amount, we reduce the computational intensity of mutual information analysis; if we extract 13 trajectories from ~150 neurons then from the naive estimate for the complexity of training Copula-GP estimators (see core paper (Kudryashova et al., 2022)) we can come to an upper-bound of $150^2 \div 13^2 \approx 133$ times faster Copula-GP training.

## 9 Bagged Copula-GP Validation Tests

To validate our bagging methodology, we tested various Copula-GP aggregation methods alongside baseline unbagged Copula-GP on generated bivariate copulas (the "true" copula). C-vines are made up of bivariate copula building blocks, with C-vine model selection essentially being made up of consecutive bivariate copula selections. As such, validation alongside baseline Copula-GP on bivariate copulas alone is indicative of improvements in C-vine entropy estimation via bagging methods described.

For all validation tests in this section, the task was to accurately replicate the true copula's parametric entropy and dependency shape, parameterized on some variable $x$. For validation, the weighted aggregation methods utilized were 1. Copulas weighted point-wise dynamically by BIC / AIC on input, 2. Copulas weighted statically by BIC / AIC on input, and 3. naive average of copulas. In addition, we also examine copula entropy point-wise root mean squared error (RMSE) of baseline and bagging methods.

To generate the data, we drew 10000 samples from the true copula, splitting samples and their corresponding parameterizations into train and test sets (80:20 split). Validation tests 1 and 2 utilized parameterization on a normally distributed variable $x \sim \mathcal{N}(0.5, 0.2)$, restricted to the interval $[0, 1]$. Validation test 3 utilized a parameterization in a time-based variable $t$ scaling linearly from 0 to 1. Copula training and model selection utilized train set samples, with accuracy validation utilizing metrics described on test set samples. All validations utilized the heuristic algorithm for individual estimator model selection, and for bagged estimations 4 estimators underwent individual training and model selection routines. We include random seed, module version, and hardware information for test reproducability purposes in the appendix (see G).

Table 1: True and predicted mean copula entropies $\overline{H}_C$ (with 95% confidence interval included in parentheses) and point-wise RMSE of validation tests. Closest to actual / best scores in bold. We find that BIC Dynamically bagged Copula-GP consistently outperforms alternative methods in accuracy.

| Model | Test 1 $\overline{H}_C$ | RMSE | Test 2 $\overline{H}_C$ | RMSE | Test 3 $\overline{H}_C$ | RMSE |
|---|---|---|---|---|---|---|
| True Copula | -0.1440 (0.0154) | - | -0.6864 (0.1148) | - | -0.6864 (0.1148) | - |
| Baseline | -0.0000 (0.0000) | 0.1442 | -0.6680 (0.0688) | 0.0694 | -0.4556 (0.2038) | 0.2587 |
| BIC Dynamic | **-0.0802** (0.0147) | **0.0647** | **-0.6764** (0.0847) | 0.0719 | **-0.5628** (0.0754) | **0.1388** |
| BIC Static | -0.0791 (0.0117) | 0.0656 | -0.6725 (0.0793) | 0.0713 | -0.5589 (0.0650) | 0.1418 |
| AIC Dynamic | -0.0796 (0.0128) | 0.0652 | -0.6664 (0.0870) | 0.0748 | -0.5382 (0.0813) | 0.1615 |
| AIC Static | -0.0790 (0.0123) | 0.0658 | -0.6662 (0.0921) | 0.0658 | -0.5389 (0.0703) | 0.1596 |
| naive average | -0.0798 (0.0133) | 0.0651 | -0.6738 (0.0778) | **0.0651** | -0.5580 (0.0722) | 0.1420 |

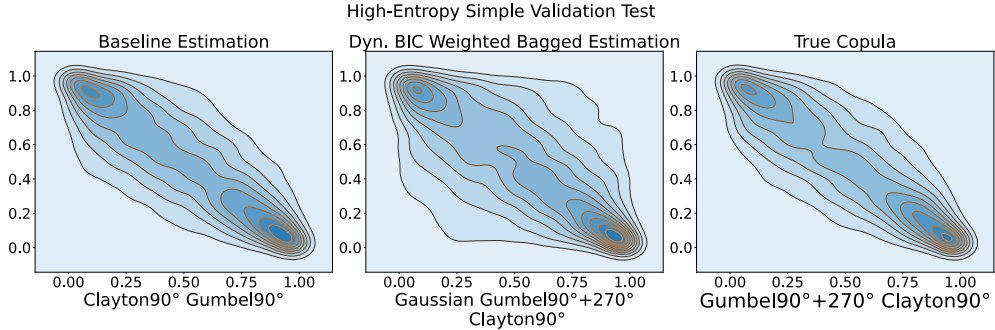

Figure 1: Average predicted copulas of BIC dynamically bagged and baseline Copula-GP on low-entropy copula validation test set, with true copula for comparison. Contour lines of copulas are visible. Note how the distribution "tricks" the baseline estimator into picking the independence copula, but the bagged estimate is able to capture major tail dependencies in the true distribution (circled in red).

## 9.1 Results

For the first validation, we expected our bagging methods to perform well. Baseline Copula-GP when facing such a copula can often "give up" early into heuristic model selection and select independence if it finds non-independence copulas possess too low WAIC. We expect that bagging is able to make up for this flaw in baseline heuristic model selection via it's bias-variance trade-off, and thus pick up tail dependencies in the model. In table 1 we observe that bagging methods all performed better than baseline (which selected independence), with BIC dynamic bagging possessing the best model accuracy and RMSE. Finally, in figure 1 we find that the BIC dynamically bagged estimated copula captures major tail distributions in the true copula, whereas the baseline estimation of independence does not.

Figure 2: Average predicted copulas of BIC dynamically bagged and baseline Copula-GP on high-entropy copula validation test set, with true copula for comparison. contour lines of copulas are visible. We find that bagging methods can result in incorrect selection of copula variant leading to possible misrepresentation of dependency shape, however the bagged estimation outperforms baseline in mean copula entropy estimated.

For the second validation test, we expected baseline to perform well and were more interested in if the bias induced by bagging can result in worse performance on distributions the baseline is well suited for. The true copula here is a high-entropy copula with simple shape, something that in the core paper (Kudryashova et al., 2022) was found to be well suited to baseline Copula-GP as a method. The BIC dynamically weighted estimate had the best estimated mean copula entropy, with the naive average estimate possessing superior RMSE. In figure 2 we find that the BIC dynamically weighted estimation included a gaussian copula and as such makes incorrect assumption surrounding the dependency shape of the true copula. Despite this, the majority of bagged copula estimates still result in similar if not slightly better performance in copula entropy prediction.

Figure 3: Average predicted copulas of BIC dynamically bagged and baseline Copula-GP on transformed high-entropy copula validation test set, with true copula for comparison. contour lines of copulas are visible. We find that the baseline estimation possesses less of a resemblance to the true copula than the bagged estimation.

The final validation was on a copula parameterized in time. In the core paper, it was found that "transformed" cases with dependency shape changing as a function in time can be more challenging for Copula-GP to predict the entropy of accurately. As such, we utilize the copula in validation test 2 and change its parameterization to be in time $t$ scaling linearly from 0 to 1. We maintain continuity when splitting into train and test set, and as such the train set consists of parameterization in $t = 0.0$ to 0.8, and the test set consists of parameterization in $t = 0.8$ to 1.0. We hope that bagging distribution estimations made by estimators trained on continuous subsets of training data can induce some robustness to time-based transformation in true dependency shape. Our hopes were confirmed, as we find in table 1 that bagging methods all find better accuracy than baseline. BIC dynamically weighted bagging results in the best mean copula entropy estimate and RMSE. From figure 3 we see that the baseline estimated copula possesses much less of a resemblance to the true copula than the bagged estimate.

### 9.2 Validation Discussion

In general, we found that bagged copula estimations utilizing 4 estimators possesses similar or better accuracy in copula entropy estimation than baseline. BIC dynamically aggregated copulas possessed the best accuracy for all validation tests conducted, often with the best point-wise RMSE. We also found that a naive average of copulas estimated performed surprisingly well, often times out performing other bagging methods. From validation test 1, we see that bagged estimates can catch dependencies in distributions that the baseline perceives as independent. In C-vine model selection, as copulas gain more conditioning marginal variables' relationships tend to more closely resemble conditional independence (Kudryashova et al., 2022). As such, we may conclude that a bagged estimator might be more likely to pick up tail dependencies in conditioned relationships where the baseline estimator might assume independence. From validation test 3 we find that the bagged estimator is more robust to transformations in dependency shape over time, and as such may yield better accuracy when predicting marginal variables' dependency shape for time-series. Finally, we note that the BIC dynamically bagged estimator does not over-estimate copula entropy in validation.

## 10 Exploration of Experimental Neuronal Dataset

For this section, the objective was to utilize Copula-GPFA to confirm high statistical dependence between the visual cortex and pupil dilation. The main data set we wished to explore is the *Visual Coding: Neuropixels* dataset, a publicly available dataset containing spike-signals recorded from the visual cortex of live mice at single neuron spatial resolution utilizing novel Neuropixel probes, a high-fidelity and -resolution brain probe developed in 2021 (Steinmetz et al., 2021). Due to the volume of *in vivo* data recorded, as well as the quality metrics included in the data, it is an invaluable tool for analysis of underlying processes within the visual cortex. All technical claims surrounding the data in question is sourced from the technical white

paper (see D). For more information surrounding data production and specificities not covered in the body of this paper, i.e specifics on how probes were used and how visual stimuli were produced, one should see the technical white paper. The spike data was collected via insertion of *Neuropixel* probes into the visual cortex of live mice. Probes record spike signals from points in space at single neuron resolution. These points are called "units," and are where neurons *might* be. Units have quality metrics attached, which can be used to filter for units of better quality (i.e. units with less noise, units which more certainly record spikes from only one neuron, etc). We chose units via the cleaning processes described in D utilizing these quality metrics, arriving at a dimensionality of 178.

Pupil dilation has been found to have close ties to processes within the visual cortex (Franke et al., 2022; Bombeke et al., 2016; Larsen et al., 2018). As this data is directly from the visual cortex, we expected a significant statistical dependence between pupil dilation and visual cortex trajectories. As such, the motivation behind utilizing this dataset is as an experimental method of model validation. We hypothesised significant information interaction extracted via Copula-GPFA, thereby effectively confirming that the visual cortex of the brain is linked to pupil dilation and agreeing with prior results in the literature (Bombeke et al., 2016; Ganea et al., 2020).

## 10.1 Runtime Breakdown

Runtime of Copula-GPFA can be broken down into three steps: transformation into latent trajectories via GPFA, fitting of multivariate distribution via Copula-GP on either part of whole data, and aggregation via BIC dynamic bagging. GPFA was fit on 100 consecutive spike trains during Drifting Gratings stimulus presentations, each of which have a uniform temporal length of 2s followed by a 1s interval during which no stimulus is presented. We fit the model for the entirety of the dataset with the dimensionality of the elbow of the log-likelihood plot (figure 9), reducing the original dimensionality of 178 down to a latent dimensionality of 13. After GPFA application, little interim processing is needed: we removed drift and concatenated trials as per the process described in section F.2 to produce semi-continuous data and mapped onto the domain $(0, 1)$ via additional interim steps described in section F.2.

We utilized Copula-GP to estimate C-vines over the concatenation of GPFA-treated trials, possessing 12 layers (78 copulas total) with parameterization in normalized pupil dilation (as per steps described in D), testing both baseline unbagged and bagged Copula-GP. For baseline, Copula-GP was fit over the full semi-continuous data of 80 trials. For bagged Copula-GP, 4 individual models were fit over 4 semi-continuous subsets of 20 trials each.

For model ensembling, we utilized the BIC dynamically-weighted aggregation method. We estimated parametric negative copula entropy $-H_C(\mathbf{X}|Y)$ over time utilizing baseline and bagged estimates for 15 continuous trials, expecting either similar or lesser (more negative) copula entropy estimates of bagged Copula-GP as baseline, as was observed in validation.

## 10.2 Results

We can see the outcome of $-H_C(X)$ and interaction information $I(X \leftarrow Y)$ calculations alongside normalized pupil dilation in figure 4. Bagged Copula-GP estimated a copula with mean entropy $-H_C(X) = 13.4027$ bits (95% CI of 0.7069) and mean $-H_C(X|Y) = 7.3323$ bits (95% CI of 5.9693). Copula entropy appears to be highly correlated with pupil dilation, with decreases in copula entropy occurring with decreases in pupil dilation. In addition, the information interaction stays firmly *negative*, implying the multivariate distribution of neuronal trajectories possesses statistical dependence on pupil dilation (thereby confirming our hypothesis).

From figure 5, we can see that like the bagged estimate the baseline negative copula entropy appears to be dependent on pupil dilation, with large dips occurring with pupil dilation. We found a mean negative copula entropy $-H_C(X)$ estimation of 13.4745 bits (95% CI of 3.0720) and a mean negative conditional copula entropy $-H_C(X|Y)$ empirical estimation of 7.2221 bits (95% CI of 6.1190), with X being the neuronal trajectories and Y being pupil dilation. However, we find the bagged estimate is less sensitive to small changes in pupil dilation than the baseline estimate. In addition, the bagged entropy estimate appears

to steadily hover around the mean negative entropy found for most observed time-buckets where as the unbagged estimate appears almost linear in pupil dilation (see figure 5). These differences aside, the bagged and baseline mean copula entropy and information interaction estimates were close, implying that the bagged estimator does not inherently out- or under-perform baseline unbagged *Copula-GPFA* in the case of the *in vivo* data.

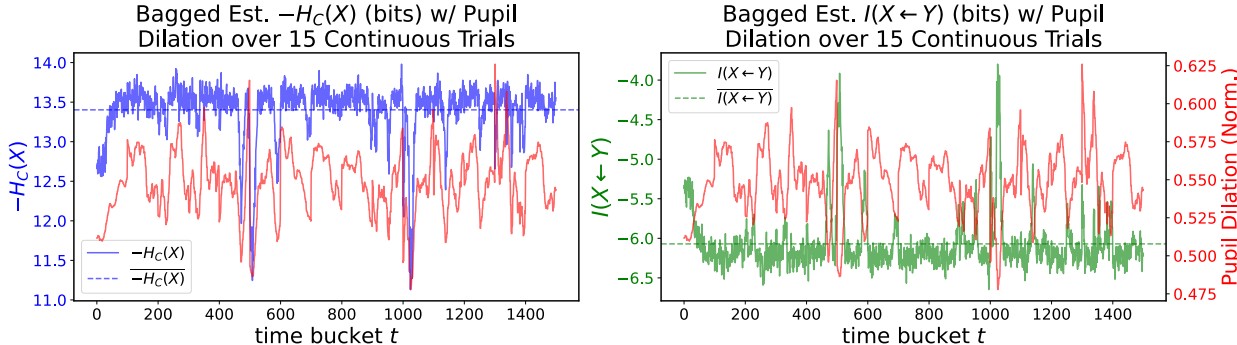

Figure 4: Pupil dilation (in red), copula entropy of neuronal trajectories parametric in pupil dilation utilizing bagged estimation (in blue), and corresponding information interaction (in green) through time utilizing estimated copula entropy. Copula entropy was estimated via estimating a distribution with BIC dynamically bagged Copula-GP.

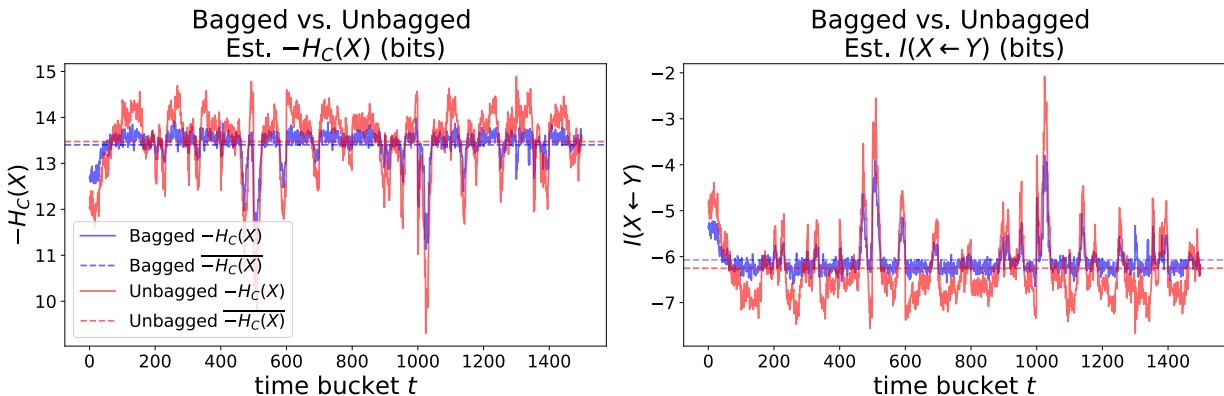

Figure 5: BIC dynamically bagged vs unbagged *Copula-GP* estimated entropies and information interactions. The BIC dynamically bagged entropies curve seems to hover steadily around a mean, and does not appear to be as volatile as unbagged entropies.

## 11 Discussion

**Copula-GPFA is able to efficiently provide accurate and meaningful dependency and information quantification analysis and results.** We validated Copula-GPFA as a mutual information and information interaction extraction method on *in vivo* data extracted from mice, confirming significantly negative information interaction between neuronal trajectories and pupil dilation and agreeing with prior findings linking the visual cortex with pupil dilation (Bombeke et al., 2016; Ganea et al., 2020; Zylberberg et al., 2012). In addition, Copula-GPFA also allows for easy interpretability of findings, allowing for visualization of both neuronal trajectories and information quantities over time. We also justify a significant reduction in computational complexity via GPFA utilized with Copula-GP, using complexity estimates made in the core paper.

**Dynamically bagging Copula-GP estimates over subsets of data provides superior estimations over baseline in tail-dependency identification and estimated copula entropy accuracy in the bivariate case.** We also introduced copula bagging as a means to possibly improve copula estimation, complete with both formalistic and practical justifications, and implemented bagging as an extension to the Copula-GP code-base (locally). In addition, we utilized the bagged extension on the experimental *in vivo* data, finding results in agreement with baseline estimation of mean copula entropy with more stability over time, however whether this stability is indicative of the true distribution or not goes unconfirmed. As Copula-GPFA is not inherently a method restricted to the area of neuroscience, it is entirely plausible Copula-GPFA is able to be meaningfully applied to other fields where dependency analysis is useful (i.e. bio-informatics or quantitative finance).

While we did not test robustness of the bagged extension to higher dimensionality, as C-vine selection within the Copula-GP algorithm consists of individual selections of bivariate building blocks we hypothesize that benefits found in the bivariate case propagate into cases of higher dimensions. Our implementation of bagging also follows in the footsteps of the formalistic approach of the original Copula-GP implementation, and as such maintains flexibility in use cases outside the realm of spike train data and neuroscience. We also acknowledge that the ability to bag copula estimations allows for the aggregation of Copula-GP estimators to be fit on individual trials of data, thereby resolving possible violations of local smoothness and continuity assumptions caused by concatenation of GPFA treated spike train trials (due to time-to-fit concerns and limitations in compute-power, this goes unutilized as of now). Finally, we note our bagging implementation can be used *without* heuristic model selection, and one could utilize our implementation to bag singleton mixtures (mixture copulas of only a single variant) as an alternative efficient means of model selection. Finally, we note that while we did not thoroughly investigate individual bivariate copulas within the C-vines estimated, *each bivariate copula itself represents a dependency relationship.* As such, there is room for more analysis.

### 11.1 Further Validation, Optimization, and Use of Bagging Methods

Our validations find that weighted copula estimate aggregation can yield effective improvements in copula tail dependency identification and copula entropy accuracy. The obvious extension to validations made would of course be to confirm if such improvements firmly carry over to higher dimensions, however because Copula-GP implementation and the structure of the C-vine we believe that benefits found in the bivariate case will propagate into higher dimensions. In addition, further refinement of aggregation methods may be considered, i.e. clustering methods such as $k$-means as a way to select copula estimates for datasets. Use of parallel processing in training bagged estimators concurrently might also be a natural optimization for bagged Copula-GP; one could even bag bivariate copulas immediately after they've completed model selection on their subsets of data during vine training.

That being said, benefits found in the bivariate case are still entirely applicable. Bivariate copulas are still effectively used in fields from computational finance (Cherubini et al., 2011) to bio-informatics (Ray et al., 2020). As such, the improvements made to Copula-GP in bivariate copula estimation via the addition of dynamically weighted aggregation methods may be used to robustify bivariate copula selection effectiveness in such use cases.

Finally, we believe there is a gap within the literature for a more comprehensive review of mutual information estimation. While Copula-GP was tested against other commonly used methods in the core paper (KSG by Kraskov et al. (2004), BI-KSG by Gao et al. (2016), and MINE by Belghazi et al. (2021)), other methods have recently been produced with robustness to high-dimensionality. In particular, several recent novel Density Ratio Estimation (DRE) based methods have been produced, such as TRE by Rhodes et al. (2020), DRE-$\infty$ by Choi et al. (2022), and MDRE by Srivastava et al. (2023).

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

## A  Machine Specifications

Experiments were ran on a machine utilizing 128 GB of system memory, an Intel Xeon Gold 6142 processor, an NVidia 2080, and an NVidia 2080Ti. Plots were made on a laptop with 8 GB of system memory and an Apple M2 processor. The python version utilized for all computations and plots made was 3.10.6; *Elephant* 1.0.0 and *Copula-GP* 0.0.5 were utilized for computations, and *Matplotlib* 3.6.1 and *Seaborn* 0.10.0 were utilized for plot production.

## B  Information Theory Essentials

Information theory, in broad terms, is the study of formal information quantification and communication (Shannon & Weaver, 1998). The key measure of information in this field is *entropy*, defined as $H(X) =$

$-\mathbb{E}_X(\log p(x)), \; x \sim X$. Essentially, entropy measures the number of *bits of information* needed to encode the total uncertainty of event $X$ (Shannon & Weaver, 1998), and as such acts as a quantification of uncertainty. We also define the conditional entropy of $X$ given the outcome $y$ of event $Y$ as

$$H(X|Y) = -\mathbb{E}_{X,Y}(\log \frac{p(x,y)}{p(y)}), \; x \sim X, \; y \sim Y, \tag{14}$$

and the *mutual information* between $X$ and $Y$ as $I(X:Y) = H(X) - H(X|Y)$. Mutual information thus represents the *reduction in bits of uncertainty* surrounding $X$ given the outcome of $Y$. In addition, this quantity is *symmetric* (Shannon & Weaver, 1998), and so $I(X:Y) = H(Y) - H(Y|X)$ as well. In other words, **mutual information describes statistical co-dependence between events $X$ and $Y$.** In addition, we note that mutual information is positive semi-definite ($\geq 0$), and so only ever represents a reduction in uncertainty (this is intuitive; the unconditioned outcome of an event naturally encompasses all conditions). We extend this to the *conditional mutual information* of $X$ and $Y$ given the outcome of event $Z$, $I(X:Y|Z) = H(X|Z) - H(X,Y|Z)$, where $H(X,Y|Z)$ is the *conditional joint entropy* of $X$ and $Y$ given the outcome of $Z$, or the bits of uncertainty in the joint system of $X$ and $Y$ under condition $z$ (Shannon & Weaver, 1998). Finally, mutual information is easily extendable to a definition of *joint mutual information* via replacing singular entropy with it's joint definition (as traditional entropy itself is not the primary concern of this paper, we will skip rigorously defining these terms for brevity) (Shannon & Weaver, 1998). Joint mutual information and it's conditional counterpart are the primary quantities of interest we wish to estimate in this paper as we wish to extract the *interaction information* between $\mathbf{X}$ and $Y$ $I(\mathbf{X} \leftarrow Y) = I(\mathbf{X}|Y) - I(\mathbf{X})$, representing the change in the mutual information between marginals of $\mathbf{X}$ when $Y$ is learned, or the amount of uncertainty surrounding $\mathbf{X}$ captured in $Y$.

## C    Python Version and Hardware Utilized

Computations were made on the Edinburgh University compute server utilizing 128 GB of system memory, an Intel Xeon Gold 6142 processor, an NVidia 2080, and an NVidia 2080Ti. Plots were made on a laptop with 8 GB of system memory and an Apple M2 processor. The python version utilized for all computations and plots made was 3.10.6; *Elephant* 1.0.0 and Copula-GP 0.0.5 were utilized for computations, and *Matplotlib* 3.6.1 and *Seaborn* 0.10.0 were utilized for plot production.

Table 2: Different GPLink functions used in Copula-GP for different copula types. GPLinks are used to parameterize dependence of the particular copula in a continuous variable. Courtesy of (Kudryashova et al., 2022).

| Copula | Domain | GPLink($f$): $\mathbb{R} \to \mathrm{dom}(c_j)$ |
|---|---|---|
| Independence | - | - |
| Gaussian | $[-1, 1]$ | $\mathrm{Erf}(f/1.4)$ |
| Frank | $(-\infty, \infty)$ | $0.3 \cdot f + \mathrm{sign}(f) \cdot (0.3 \cdot f)^2$ |
| Clayton | $[0, \infty)$ | $\mathrm{Exp}(0.3 \cdot f)$ |
| Gumbel | $[1, \infty)$ | $1 + \mathrm{Exp}(0.3 \cdot f)$ |

## D    Data Retrieval and Cleaning

*Visual Coding - Neuropixels* is split into recording sessions, where during each session a mouse test subject was exposed to varying visual stimuli, and is publicly available via the `allensdk` python package (visit `https://portal.brain-map.org/` and navigate to "circuits and behavior", then "Neuropixels"). Stimuli were presented in "blocks" of similar kinds of stimuli of varying duration. For this project, we examine specifically the first 100 stimuli presentations of the first "drifting gratings" block (block 2), as this way we mitigate variation in neuronal response due to differing stimulus length. In addition, within this block individual presentations are of uniform length (2 seconds of presentation, with a inter-presentation break of 1 second), and so the block is easily divisible into separate spike train trials (the GPFA implementation used is best suited for a trial-by-trial format).

The session data contains the full spike recordings of all recorded units within the subject's visual cortex. As stated prior, these units possess quality metrics that correspond to how accurate and noisy unit recordings are. Of these, we utilize Signal to Noise Ratio (SNR) and Inter Spike Interval (ISI) Violation rate. SNR corresponds to the ratio of the maximum unit waveform amplitude to one standard deviation of the waveform, and acts as a metric of how noisy the unit recording is (Biau & Scornet, 2016). ISI violation rate is the percentage of unit spikes that occur during what should be the corresponding neuron's refractory period, and serves as a metric to determine whether multiple neurons and/or electronic interference are being picked up in a single unit's recordings (Czanner et al., 2015).

Session data used here was pulled from a single session (session ID 756029989). By setting a lower bound for SNR and an upper bound for ISI violation rate, we are able to filter overly noisy and unreliable units' spike recordings out of the data set. Utilizing a SNR lower bound of 3 and an ISI violation rate upper bound of 0.05 (5%), we found that in this session 26.0% of units meet these quality thresholds (178 out of the original total of 684). Example spike data for a single stimulus presentation of 2 seconds is shown in figure 11.

We also utilized pupil area recordings ($cm^2$) to parameterize the copulas in pupil dilation. Measurements were recorded every 33ms and possess a relatively large range, with sporadic large spikes in pupil dilation (see pupil dilation curves in figure $10a$). As such, we applied a rolling mean with a window of 10 entries for smoothing followed by a robust normalization procedure $\text{Robust}(X)$:

$$\text{Robust}(X) = \frac{X - Q_1(X)}{Q_3(X) - Q_1(X)}, \tag{15}$$

where $Q_1(X), Q_3(X)$ are the 1st and 3rd quartile values of $X$. Doing so, we reduce the impact of outliers in the data. As the input for parameterizing values for fitting a C-vine via Copula-GP must be the interval $[0, 1]$, we map onto via an additional min-max normalization

$$\text{MinMax}(X) = \frac{X - \min(X)}{\max(X) - \min(X)}. \tag{16}$$

We can observe the difference between the raw and processed data in figure 10. See figure 10c for the distribution this preprocessing regime created, which appears roughly normal with few outliers.

The technical white-paper for the dataset is also available at:
https://brainmapportal-live-4cc80a57cd6e400d854-f7fdcae.divio-media.net/filer_public/80/75/8075a100-ca64-429a-b39a-569121b612b2/neuropixels_visual_coding_-_white_paper_v10.pdf.

# E Copula-GP Implementation

The vine construction implemented in Copula-GP used copula building blocks from five distinct families: independence, Gumbel, Gaussian, Frank, and Clayton copulas, with the independence copula preferred for independent variables (see figure 6). The factorization of the vine is that of the C-vine described by equation 3, with accommodations made for GP-parameterization. To fully capture the tail dependencies and negative correlations in relationships between marginal distributions, *mixed copulas* were utilized. In the core paper, these are defined as

$$C_{mixed}(\mathbf{X}|Y) = \sum_{j=1}^{K} \phi_j(Y) C_j(\mathbf{X}|\theta_j(Y)), \tag{17}$$

where $K$ is the number of elements, $\phi_j$ is the *concentration* of the $j$th copula ($c_j$), and $\theta_j$ is the $j$th copula's parameter GPLink. The GPLink for each copula is determined by it's copula variant, as shown in figure 2, with $\theta$ being defined by $\text{GPLink}(f)$, where $f$ is sampled from $\theta \sim \mathcal{N}(\mu, K_\lambda(X, X))$ (the choice of GPLink depends on the kind of copula; see 2). GP is also utilized to parameterize the concentrations $\phi_j$, which are defined as

$$\phi_j = (1 - t_j) \prod_{m=1}^{j-1} t_m, \quad t_m = \Phi\left(\tilde{f}_m + \Phi^{-1}\left(\frac{M - m - 1}{M - m}\right)\right), \quad t_M = 0, \tag{18}$$

where $\Phi$ is the CDF of a standard normal distribution and $\tilde{f}_m$ is sampled from $\tilde{\mathbf{f}}_{\mathbf{m}} \sim \mathcal{N}(\tilde{\mu}_m, \ \tilde{K}_{\tilde{\lambda}_m}(Y,Y))$. This gives us $2M-1$ sets of hyper parameters to estimate, $\{\lambda\}_M$ kernel hyperparameters for each GPLink $\theta$ and $\{\tilde{\lambda}\}_{M-1}$ kernel hyperparameters for each concentration function $\phi$, estimated via the methods described in section E.1.

### E.1  Copula-GP Model Selection and Parameter Estimation

As stated in the body of this paper, the parameters for the distributions used in GP must be estimated. In the Copula-GP framework, this is accomplished through the use of *stochastic variational inference (SVI)*, with SVI being scaled to high dimensions by means of *Kernel Interpolation for Scalable Structured Gaussian Process (KISS-GP)* (Wilson & Nickisch, 2015). These methods were specifically used for efficient implementation in aforementioned python libraries *PyTorch* and *GPyTorch* (Paszke et al., 2019; Gardner et al., 2018). For model hyper-parameter selection, *Watanabe-–Akaike information criterion (WAIC)* was used, a metric which aims to maximise the Akaike information criterion (AIC) by means of a Bayesian-approach (we more rigorously define AIC in section 7). Given $f_1$, $f_2$ and the true distribution $g$, WAIC examines the difference in log-likelihood i.e. $I(g:f_1) - I(g:f_2) = -\mathbb{E}(\log f_1(X) - \log f_2(X))$, and selects the model with the greater mutual information (Watanabe, 2012). The space of models is too large to find the optimal model when considering the number of combinations of copulas shown in 6, and as such the core paper implements a greedy algorithm of minimising WAIC which can be used with all copula types and a heuristic algorithm specifically tuned for certain combinations of copula types. For this project, we utilize the heuristic approach of copula selection.

## F  GPFA Specifics

### F.1  Dimensionality Selection

Before we can utilize GPFA, we first must isolate which number of dimensions $n$ to reduce down to. For the *in vivo* dataset we chose $n$ via investigating the log likelihood of GPFA fits extracted from a 3-fold cross validation, from target dimensionality $n = 1$ to 50. We then plotted the log likelihoods and saw both where the elbow, or the point of maximum concave slope curvature representing a "good enough" dimension to reduce down to (Antunes et al., 2018) of the log likelihood curve was. While there are computational methods for elbow selection (Antunes et al., 2018), we investigated few enough points to allow for the elbow to be selected ourselves visually.

### F.2  Additional Post-GPFA Interim Processing Required

Copula-GP's C-vine framework is fit on single-trial continuous, however *Elephant*'s GPFA implementation produces trajectory data that is split into trials. A solution to this would be to concatenate trials trajectory wise, however per-trial drift in trajectory means can result in weak Copula-GP fit performance if these lead to large jump discontinuities and thus loss of smoothness in the data; the kernel for the GP-link functions used in parameterization will require more restrictions as it encodes the smoothness of the data (among other things) (Schulz et al., 2018; Yu et al., 2009). For the *in vivo* data, we found in figure 7 that drift occurs in the 1 second inter-stimulus break in the average trial. As such, we crop this period out of each trial (50 points), and concatenate trials together trajectory-wise. While this is by far not a perfect solution, it allows the data to remain roughly smooth at the cost of continuity in time and residual (small) jump discontinuities, as well as isolating the data to only when stimulus presentations are occurring. See figure 8 for a example of trial-to-trial discontinuities created by this interim step.

### F.3  Python Implementation of GPFA Used

The implementation of GPFA utilized is sourced from the *Elephant* (Electrophysiology Analysis Toolkit) python library, with the package's 1.0.0 release (used for the contents of this paper) being published November 10, 2023 (Denker & Kern, 2023). The *Elephant* package was specifically designed for use on neuronal data, motivated by a push to release a standardized python package for use in computation neuroscience. The

GPFA module has specifically seen use in recent papers (Pei et al., 2022; Bagi et al., 2022), and since release has become quite popular. The GPFA module receives some dimension $n$ to reduce down to and time-bucket size $m$ (we utilize 10ms and 20ms buckets) to instantiate a GPFA python object. This object can then be fit on a number of spike train recordings of uniform temporal length given the start and end time of each recording, summing the number of spikes for each time bucket and utilizing the bucketed spike counts as the observation matrix in equation equation 13. As each projection corresponds to the expectation of the latent trajectories $\mathbb{E}(\Theta|\Psi)$, these parameters are estimated in the *Elephant* implementation via expectation maximization (Yu et al., 2009).

---

**Algorithm 1** Implemented bivariate mixture copula bagging algorithm. Here, mixture copulas consist of mixed copula variants *variants*, their corresponding dependency parameter values $\Theta$ (for each of the $m$ inputs), and their corresponding mixing parameters *mix*. We assume weights have been defined over all $m$ input points.

---

1: **Input**
2:   *mixture*: A list of mixture copulas to aggregate.
3:   *weights*: Corresponding mixture weights.
4: **Output**
5:   *mixture(Variant List, Mix Parameters, $\Theta$)*: A new mixture copula of variants *Variant List* with corresponding mixture parameters *Mix Parameters* and dependency parameters $\Theta$.
6: **procedure** BAGCOPULAS(*mixtures, weights*)
7:   $N \leftarrow 0$                 ▷ Unique variant counter.
8:   *indexes* $\leftarrow$ Empty dictionary         ▷ Indexing dictionary.
9:   *Variant Counts* $\leftarrow$ Empty dictionary      ▷ Counting dictionary.
10:   *Total Variant Weight* $\leftarrow$ *dict()*       ▷ Weighting dictionary.
11:   **for** each $i$-th mixture copula in *mixtures* **do**
12:    **for** each $n$-th variant in the mixture copula **do**
13:     **if** the variant has not yet been encountered **then**
14:      Let the variant act as a key for $N$ from *indexes*
15:      Let the $N$-th index of *Variant Counts* get 0
16:      Let the $N$-th index of *Total Variant Weight* get 0
17:      Increment $N$ by 1
18:     $idx \leftarrow$ corresponding entry for the variant in *indexes*
19:     Increment the $idx$-th entry of *Variant Counts* by 1
20:     Let $(i, n)$ act as a key for $idx$ from *indexes*
21:     Add the the $i$-th entry of *weights* to the $idx$-th entry of *Total Variant Weight*
22:   *Variant List* $\leftarrow$ an empty list
23:   *Mix Parameters* $\leftarrow \mathbf{0}_{(N \times m)}$         ▷ Mixture parameters.
24:   $\Theta \leftarrow \mathbf{0}_{(N \times m)}$           ▷ Dependency parameters.
25:   **for** each $i$-th mixture copula in *mixtures* **do**
26:    **for** each $n$-th variant in the mixture copula **do**
27:     $idx \leftarrow$ corresponding entry for $(i, n)$ in *indexes*
28:     Set the $idx$-th entry of *Variant List* to the variant
29:     $weight \leftarrow$ the $i$-th entry of *weights*
30:     $\gamma \leftarrow$ the $n$-th mixture parameter of the mixture
31:     Add $\gamma \times weight$ to the $idx$-th row of *Mix Parameters*
32:     $\theta \leftarrow$ the $n$-th dependency parameter of the mixture
33:     *Total Weight* $\leftarrow$ the $idx$-th entry of *Total Variant Weights*
34:     Add $\theta \times weight / Total\ Weight$ to the $idx$-th row of $\Theta$
35:   **return** A new mixture copula *mixture(Variant List, Mix Parameters, $\Theta$)*

---

# G  Validation Test Reproducability

All validation tests were done using variations of arguments for the `validate_bagging.py` python file.

**Test 1**  For test 1, we utilized:

- Random Seed: 859448723
- Num. Estimators: 4
- Maximum Copula Elements: 5
- Dimensions: 2
- Shuffling of Data: Yes
- Input Type: Random

**Test 2**  For test 2, we utilized:

- Random Seed: 859443
- Num. Estimators: 4
- Maximum Copula Elements: 3
- Dimensions: 2
- Shuffling of Data: Yes
- Input Type: Random

**Test 3**  For test 3, we utilized:

- Random Seed: 859443
- Num. Estimators: 4
- Maximum Copula Elements: 3
- Dimensions: 2
- Shuffling of Data: No
- Input Type: Linear

# H   Additional Figures

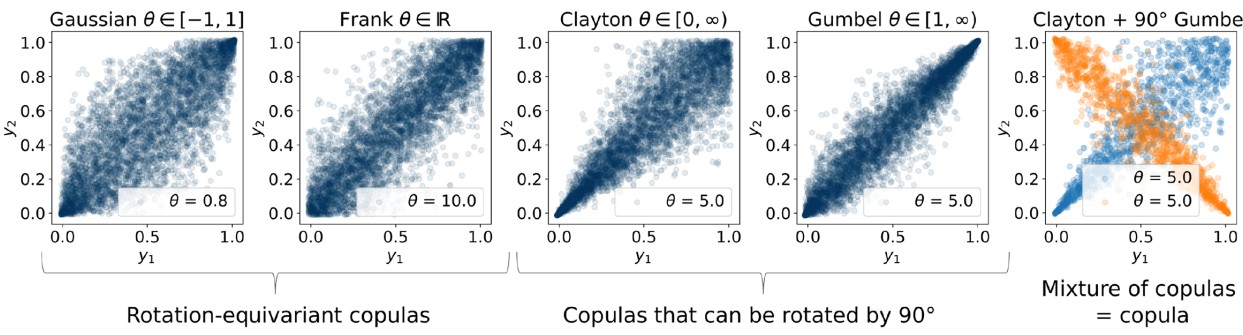

Figure 6: Various kinds of copulas utilized in Copula-GP. Note the difference in tail distribution representation, as well as how combining copula variants make a new *mixed* copula. Original source (Kudryashova et al., 2022), with permission.

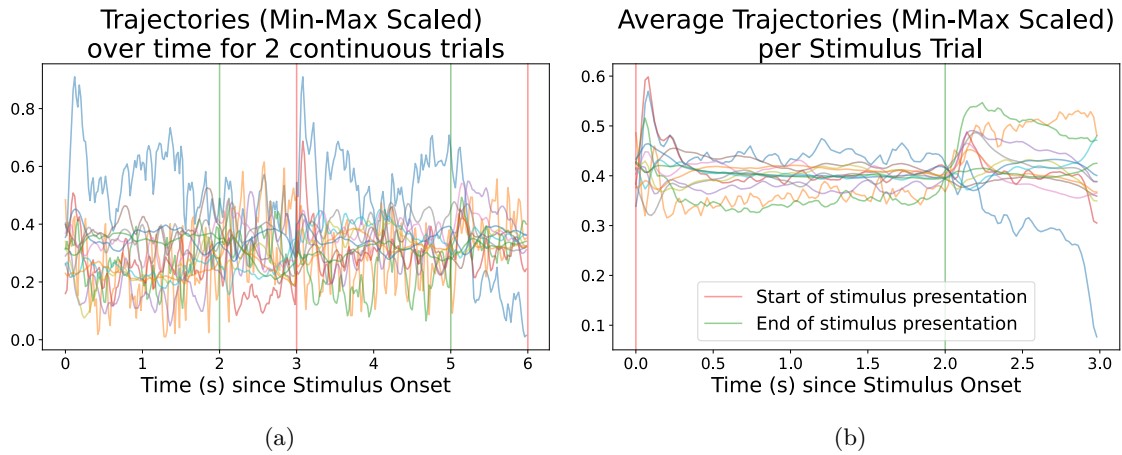

Figure 7: Trajectories extracted via GPFA (dimension $n = 13$) representing dynamics driving recorded neuronal activity. Figure 7a are two trials' trajectory data as functions in time that have been concatenated, giving the appearance of continuity. Figure 7b is the mean trial. Note the clear vertical drift in mean and single trial trajectories post-stimulus presentation stop. Data is scaled to the range [0.01,0.99] to match Copula-GP's input range of (0,1). Lines are transluscent for visibility purposes.

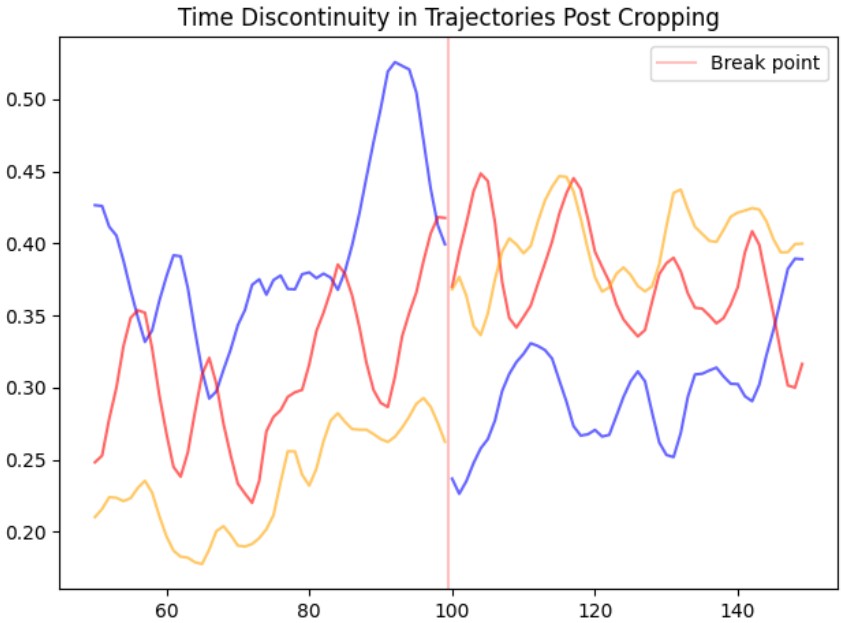

Figure 8: Discontinuities present in trajectory data post-cropping and -concatenation (3 of 13 trajectories shown). Such discontinuities can negatively effect GP performance when estimating GP-link functions during the Copula-GP fit process.

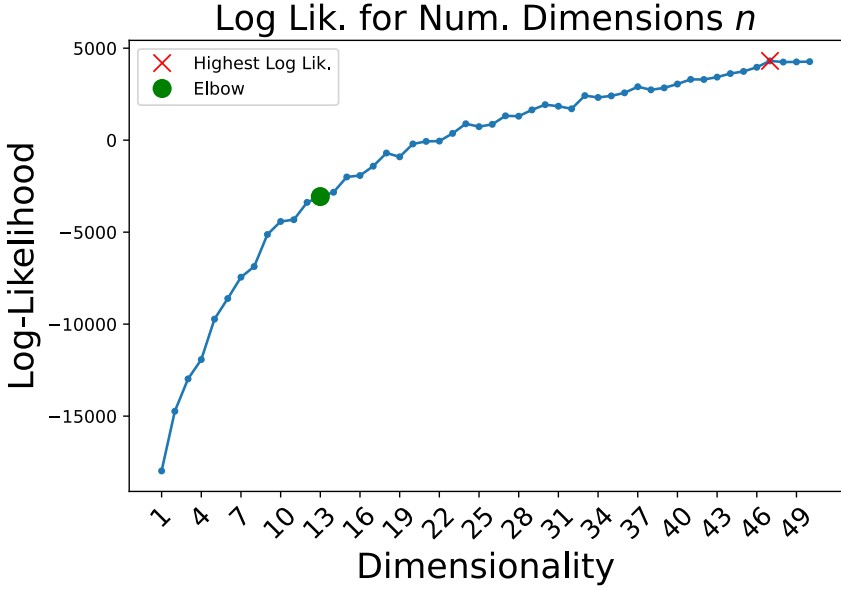

Figure 9: Log-likelihood curve for GPFA fit on trials as function of dimension $n$. Elbow and max log-likelihood found marked. Note that only dimension up to and including $n = 50$ were tested, and it is entirely possible log-likelihood to increase further in dimensions.

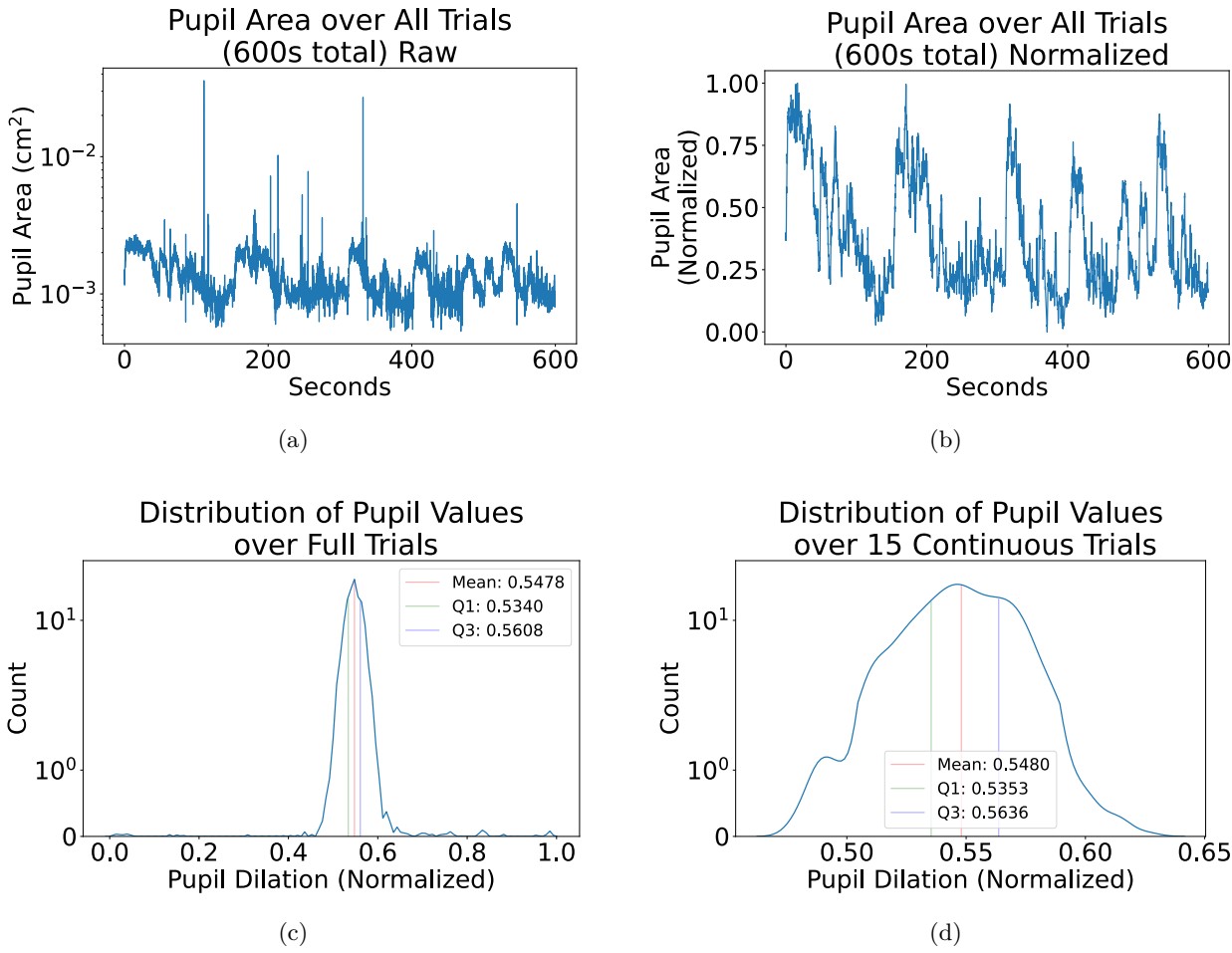

Figure 10: Figures 10a and 10b represent pupil dilation as function of time over all trials (duration 600s), raw (10a, pupil area in cm$^2$ scaled logistically) and processed (10b, pupil area smoothed and normalized). Processing consisted of rolling mean (window of 10 entries), a robust scaling, and a min-max scaling. Note the removal of strong outliers through processing, as well as the large range necessitating a logistic scale in values present in the raw recordings. Figure 10c represents the distribution over all pupil dilation data utilized (100000 continuous samples / 100 trials). Figure 10d represents the distribution over 1500 continuous samples (15 trials). Note the use of logistic scale in figures 10a, 10c, and 10d.

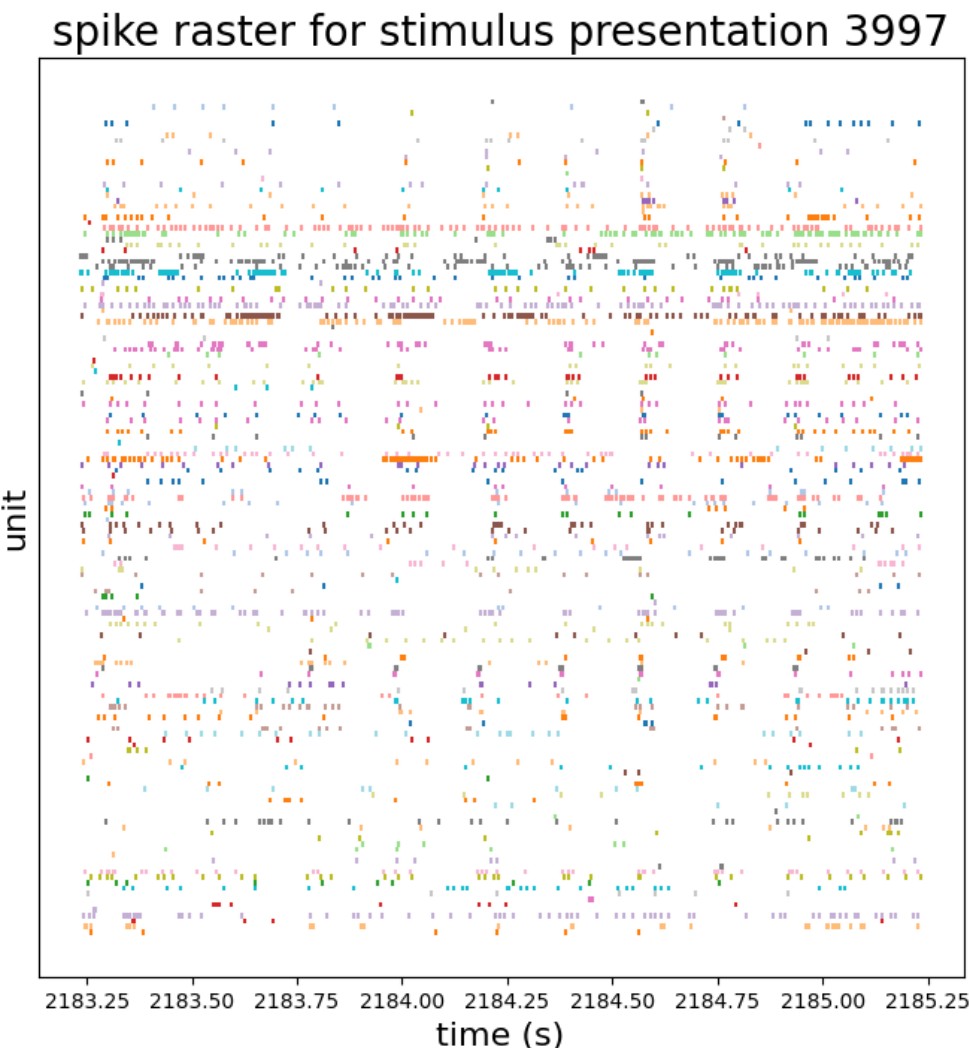

Figure 11: Spike raster of a single drifting gratings stimulus presentation (duration of 2s), from the session utilized in model validation (session ID 756029989). Only those spikes with SNR lower bound of 3 and ISI violation rate upper bound of 0.05 are shown. Note correlation in spike events present in the raster plot, as well as heightened neuronal spike-rate at the beginning of stimulus presentation.

