# OpenReview forum: "Application of Bagged Copula-GP: Confirming Neural Dependency on Pupil Dilation"
_TMLR — Accepted by TMLR_

### Review · Reviewer_paFS · 2024-09-22

**Summary Of Contributions:**

The presented work introduces two additions on top of Copula-GP to improve the estimation of the dependency between a set of variables and tested on neuronal data.

One of the proposed additions is combining GPFA with Copula-GP. GPFA is a dimensionality reduction strategy that helps the Copula-GP to perform in a lower dimensional space.

The other proposed addition is using bagging in combination with Copula-GP. Bagging is a bootstraping strategy in which several 'weak' models are trained and the prediction of all models is combined to provide a stronger prediction. The authors propose learning multiple copulas and provide the estimated copula as a the mean of the mixture.

**Audience:**

Yes

**Broader Impact Concerns:**

None that I am aware.

**Claims And Evidence:**

Yes

**Requested Changes:**

- Improve Structure of the paper: Adapt the paper to have less larger sections, rather than a lot of short sections.

**Strengths And Weaknesses:**

**Strenghts**
- The paper presents an extensive experimental section in which the influence of the proposed additions to Copula-GP are evaluated in contrast with the lack of them.
- The paper provides sufficient background information on what a Copula is, what Copula-GP is to make the paper easier to follow to readers that are not familiar with the topic.

**Weaknesses**
- The presented work looks more like a set of proposed additions over Copula-GP to improve its performance rather than a simple clear idea.
- The dimensionality of the data is not very high.
- The paper's structure has too many sections, some of them being very short. The paper would benefit from a re-structuring, by adding more subsection and reducing the number of sections.

---

> ### Author Response · Authors · 2024-10-16
> **Authors' Response**
>
> We thank you for your review, and would like to address how our revision has affected some of the weaknesses and requested changes noted:
> - **Weakness 2 - Dimensionality of the Data:** We have added the original dimensionality of the experimental dataset (178).
> - **Weackness 3 / Requested Change - Number of Sections:** We have reduced the number of sections and subsections utilized.
>
> Additional note: We have moved much of the formalism to the main body, as per Reviewer CXvm's request. We suggest you read the revision's sections 4, 5, and 6, which have been re-hauled.
>
> Please let us know if you have any other thoughts! Best regards,
>
> The Authors.

---

### Review · Reviewer_CXvm · 2024-10-08

**Summary Of Contributions:**

This paper introduced a framework of ensemble multiple copula estimators, with the goal of estimating mutual information/information interaction between groups of brain's neurons conditional to behavioral variable. In particular, the authors proposed to combine bagging methods with dimension reduction using Gaussian Process Factor Analysis (GPFA) and Copula-GP, and show with numerical validations on simulated and realistic (neuronal data of visual cortex on pupil dilation) datasets that this combination of frameworks help to improve the performance of the mutual information estimation task.

**Audience:**

No

**Claims And Evidence:**

Yes

**Requested Changes:**

-  I suggest the authors to move many parts of the formalism in the Appendix to background sections 4-5-6.
- Math typo in the first paragraph of Section 6: mutual information $I(X, Y) = H_C(X) - H_C(X \mid Y)$, not the other way around.
- I am aware of two of the works that can be used to estimate Mutual Information that can scale to high-dimensional data: TRE (Rhodes et al. 2020) and DRE (Choi et al. 2022). Could the authors perform the mutual information estimation task with high-dimensional Gaussians in Section 6.2 of Choi et al. (2022) with their framework included?
- Could the authors provide the runtime of the methods, separated by each step (GPFA, fitting copula, ensembling)?

Rhodes, Benjamin, Kai Xu, and Michael U. Gutmann. "Telescoping density-ratio estimation." Advances in neural information processing systems 33 (2020): 4905-4916.

Choi, K., Meng, C., Song, Y., & Ermon, S. (2022, May). Density ratio estimation via infinitesimal classification. In International Conference on Artificial Intelligence and Statistics (pp. 2552-2573). PMLR.

**Strengths And Weaknesses:**

## Strengths
- Motivation and Background: the paper discusses the challenges in estimating the statistical dependence in neuronal data, where it is typical to deal with high-dimensional observations.
-  The empirical validations are sound and help demonstrate the effectiveness of the proposed method.

## Weaknesses
- The paper needs a major revision on the writing: technical background sections (Section 4-5-6) need to have more rigorous formalism -- for the moment they are too verbose; citations get duplicated in different places, and almost all of the citations are done in the wrong way (need to use \citep); the authors should break large paragraphs into smaller ones. Overall, in my opinion, the writing and formalism, in the current version, do not match the usual standard of a Machine Learning venue.
-  The novelty is quite limited. I believe all of the core methodologies in the papers are taken from previous works. For example, the proof of mutual information equals copula entropy is the same as in the original paper that introduced them (Ma and Sun 2011). The authors, however, did indeed state that their submission is a "validation of combining Copula-GP with GPFA", adding the bagging as the ensemble.
- I have a concern about the high dimensionality of the realistic dataset benchmark in Sections 9 and 11: in the main text, the original dimensions of the problem were not mentioned.
- The runtime of the proposed methods is not mentioned in the main text. The method might be effective, but if the runtime is too prohibitive, its practicality will remain in question.
- Nitpick: algorithm 1 is written in a pseudo-python style. It should be more generally written for readers who use other programming languages in data science/Machine Learning to understand (such as R or Julia).

---

> ### Author Response · Authors · 2024-10-16
> **Authors' Response**
>
> We sincerely thank you for your review, from which it is evident that you have taken the time to garner a deep understanding of the paper! We now wish to address some of the weaknesses and requested changes, and have provided a few questions in order to clarify a few of the requested changes:
>
> **Weaknesses and Requested Changes**
> - **Weakness 1 / Requested Change 1 - Formalism and Writing:** We sincerely apologize for not meeting the standard's of writing appropriate for such an established venue, and hope the recent revision has addressed some of your concerns with regards to this point. We hope that the recent revision has met your expectations, and we look forward to any feedback you might have!
> - **Weakness 3 - Original Dimensionality of Data:** We have addressed this and included the original dimensions of the realistic dataset utilized (178 features).
> - **Weakness 4 / Requested Change 4 - Runtime of the Methods:** We have slightly expanded on the runtimes of the different steps of our method, and will add the actually time-to-run each step and each test in a revision within the next week. Aside from this, we would like to ask what you think should be mentioned about runtime to highlight practicality? Part of this paper is about how the methods described might be applied, and as such we would like to make sure to touch each base with respect to practicality.
> - **Requested Change 3 - DRE based MI estimation:** We are unsure if we could produce the results of such a test in a timely manner due to time and workload constraints, however we will try to. We acknowledge a need within the literature to compare novel methods, and have included such in the discussion section of the recent revision; aside from TRE and DRE-inf, other novel methods have been developed since the original publishing of the Copula-GP paper, such as MIND by [Samo, 2021](https://doi.org/10.48550/arXiv.2102.13182) and MDRE by [Srivastava, 2023](https://doi.org/10.48550/arXiv.2305.00869).
> - **Requested Change 2 - Mutual Information Typo** Applied! We have swapped the terms around.
>
> **Additional Notes**
> - The pseudo-code algorithm has been moved to the appendix. We will update the pseudo-code in an upcoming revision to be more general as well. *Edit:* We have updated the pseudo-code to be more general in a recent revision.
> - As per Reviewer paFS's request, we have reduced the number of sections from 12 to 11, and reduced the number of subsections.
>
> We again thank the reviewer for their in depth review, and are on standby for any other feedback the reviewer might provide on the above or on the recent revision. Best regards,
>
> The Authors.

---

### Review · Reviewer_172m · 2024-10-08

**Summary Of Contributions:**

This paper applies existing techniques that shed light on the strengths and weaknesses of Copula Gaussian process (GP) based methods for pupil dilation dependence on visual cortex trajectories. They include results on toy dataset that show dynamically aggregated copulas via Bayesian Information Criterion (BIC) maintain competitive performance while reducing computational costs.

**Audience:**

Yes

**Claims And Evidence:**

Yes

**Requested Changes:**

* Paper readability would benefit from using `\citet` and `\citep` latex commands.
* In Hu & O’Hagan (2021), $N$ is used to represent the $|X|$ whereas in the current work $N$ is the number of Monte Carlo (MC) samples. Consider using $S$ to represent the number of MC samples.
* Axes for Figures 1, 2, and 3 would be helpful.
* I would like to see the truth or mean plotted for Figures 4 and 5.

**Strengths And Weaknesses:**

Strengths:
* The paper is well written.
* The paper includes a simple introduction to a Coupla-GP and clear explanation of bagging.
* The paper is a comprehensive application of a bagged Copula-GP (and baselines) for pupil dilation dependence on visual cortex trajectories. I especially appreciate the naive average.

Weaknesses:
* "GP is in essence a regression attempting to estimate a relationship between variables by means of Bayesian inference." This statement is significantly over simplified. A GP can be used for regression but is not a regression. I would also suggest that you cite the GP textbook here (Rasmussen & Williams, 2006).
* There is little discussion of why there is a need for an estimate approach to Copula-GP. "Computing this directly via integration over the estimated copula density is computationally expensive due to nested integrals." A runtimes comparison for both approaches should be included (at least for the toy dataset) with a more detail discussion of infeasibility for the *Neuropixels* dataset.
* I would like to see standard deviations for results in Table 1. It is unclear how important a difference in the thousandth place is without them.
* Input and label details for the *Neuropixels* dataset could be clearer.

Carl Edward Rasmussen and Christopher KI Williams. *Gaussian Processes for Machine Learning*. The MIT Press, 2006.

---

> ### Author Response · Authors · 2024-10-15
> **Author Reponse**
>
> We sincerely thank you for your time and constructive comments. We appreciate your comments on the strengths and weaknesses of our paper, as well as the requested changes you provided. We wish to address some of the stated weakness and requested changes:
>
> **Weaknesses Addressed**
> - **Regarding weakness 1 - Definition of Gaussian Process (GP):** We especially thank the reviewer for this stated weakness, which we would have missed if not for the insightful comment! We have amended that specific section in the recent revision (as well as expanded on how GP is used in the body of the paper; see first of Additional Notes).
> - **Regarding weakness 2 - Direct Integration of Mutual Information:** We have included why we utilize a Monte Carlo estimated approach; the Copula-GP package includes built in Mutual Information estimation via direct estimation, however in testing our machine often ran out of memory on GPU. As such, we estimated the entropy via sampling from the copula as described in the paper.
> - **Regarding weakness 3 - Standard Deviations in Table 1:** We have included the standard deviations of the estimated entropies in the table.
>
> **Requested Changes**
> - **Regarding change 1:** Applied! We now use `\citep` when the citation is not a part of the sentence itself.
> - **Regarding change 2:** Applied! We now use $S$ when referring to the number of samples.
> - **Regarding change 3:** Applied! We have included axes with corresponding cumulative probabilities as labels.
> - **Regarding change 4:** Semi-applied. There is no ground truth for the neuropixels data set / the trajectories themselves. We have however added the mean estimated entropies.
>
> **Additional Notes**
> - As per Reviewer CXvm's request, much of the formalism in the appendix has been moved to sections 4, 5, and 6 (which cover copulas, Copula-GP, and Entropy estimation); these sections are very different form their original states, and we encourage the reviewer to read these sections in the latest revision.
> - As per Reviewer paFS's request, we have combined several sections / subsections, and as such the number of sections has changed.
> - The pseudocode for the aggregation algorithm has been moved into the appendix to maintain the 12 page limit.
>
> We will upload a revision in the future with run times for experiments made. We again thank the reviewer for their insightful review, and are on standby in case the reviewer has any other comments on what we have stated above or the recent revision!
> Best regards,
>
> The Authors.

---

### Author Response · Authors · 2024-10-31
**A Message from the Authors**

Hello all, with the deadline for decisions fast approaching we would like to ask the reviewers to please provide any more constructive feedback if they feel it may improve the paper, particularly with regards to the revisions made with reviewers' critiques in mind! We profusely thank the reviewers for their help in providing constructive feedback, and appreciate the time taken in reading our paper.

Best regards,
The authors.

P.S. Happy Halloween!

---

### Decision · Action_Editor_kn5T · 2024-11-20

**Recommendation:** Accept with minor revision

**Comment:**

The paper is found to be sound, with a relatively narrow scope.

* Please share openly the code.
* it's copula variante -> its copula variant
* "As such, in addition to a naive average, we may choose to weigh models based off various Bayesian criterion." -> "As such, in addition to a naive average, we may choose to weigh models based on various Bayesian criterion."
* "it’s parameterization" -> "its parameterization"
* "catch dependencies in distributions that the baseline perceives as independence." -> "catch dependencies in distributions that the baseline perceives as independent."

**Audience:**

The paper addresses an old classical statistical problem, quite different from machine learning perspectives.
The estimation of Copula entropy is only used in neuroscience AFAIK.  The scope is thus quite narrow.

**Claims And Evidence:**

The paper does not bring strong novelties, but it is found to introduce a sound approach.
There is no novel derivation, but the factor analysis model and the  bagging strategy included for estimation are a relevant contribution.
The experiments are found to be  comprehensive enough.